# Epidemiological Predictors of Positive SARS-CoV-2 Polymerase Chain Reaction Test in Three Cohorts: Hospitalized Patients, Healthcare Workers, and Military Population, Serbia, 2020

**DOI:** 10.3390/ijerph20043601

**Published:** 2023-02-17

**Authors:** Vesna Šuljagić, Danijela Đurić-Petković, Srđan Lazić, Jovan Mladenović, Bojan Rakonjac, Dolores Opačić, Nenad Ljubenović, Biljana Milojković, Katarina Radojević, Ivana Nenezić, Nemanja Rančić

**Affiliations:** 1Department of Healthcare-Related Infection Control, Military Medical Academy, 11000 Belgrade, Serbia; 2Medical Faculty, Military Medical Academy, University of Defence, 11000 Belgrade, Serbia; 3Institute of Microbiology, Military Medical Academy, 11000 Belgrade, Serbia; 4Institute of Epidemiology, Military Medical Academy, 11000 Belgrade, Serbia; 5Torlak Institute of Virology, Vaccines, and Serums, 11000 Belgrade, Serbia; 6Centre for Clinical Pharmacology, Military Medical Academy, 11000 Belgrade, Serbia

**Keywords:** healthcare workers, military personnel, SARS-CoV-2 test

## Abstract

(1) Background: Severe acute respiratory syndrome coronavirus 2 (SARS-CoV-2) and its resulting coronavirus disease 2019 (COVID-19) has caused a fast-moving pandemic. Diagnostic testing, aimed to identify patients infected with SARS-CoV-2, plays a key role in controlling the COVID-19 pandemic in different populations. (2) Methods: This retrospective cohort study aimed to investigate predictors associated with positive polymerase chain reaction (PCR) SARS-CoV-2 test results in hospitalized patients, healthcare workers (HCWs), and military personnel (MP) during 2020, before the widespread availability of COVID-19 vaccines. Persons with a positive test result were compared with persons with a negative test result in three cohorts during the study period. (3) Results: A total of 6912 respondents were tested, and 1334 (19.3%) of them had positive PCR SARS-CoV-2 test results. Contact with a known COVID-19 case within 14 days (*p* < 0.001; OR: 1.48; 95% CI: 1.25–1.76), fever (*p* < 0.001; OR: 3.66; 95% CI: 3.04–4.41), cough (*p* < 0.001; OR: 1.91; 95% CI: 1.59–2.30), headache (*p* = 0.028; OR: 1.24; 95% CI: 1.02–1.50), and myalgia/arthralgia (*p* < 0.001; OR: 1.99; 95% CI: 1.65–2.42) were independently associated with positive PCR SARS-CoV-2 test results in the cohort of MP. Furthermore, fever (*p* < 0.001; OR: 2.75; 95% CI: 1.83–4.13), cough (*p* < 0.001; OR: 2.04; 95% CI: 1.32–3.13), headache (*p* = 0.008; OR: 1.76; 95% CI: 1.15–2.68), and myalgia/arthralgia (*p* = 0.039; OR: 1.58; 95% CI: 1.02–2.45) were independently associated with positive PCR SARS-CoV-2 test results in the cohort of HCWs. Moreover, independent predictors of positive PCR SARS-CoV-2 test results in hospitalized patients were contact with a known COVID-19 case within 14 days (*p* < 0.001; OR: 2.56; 95% CI: 1.71–3.83), fever (*p* < 0.001; OR: 1.89; 95% CI: 1.38–2.59), pneumonia (*p* = 0.041; OR: 1.45; 95% CI: 1.01–2.09), and neurological diseases (*p* = 0.009; OR: 0.375; 95% CI: 0.18–0.78). (4) Conclusions: According to data gathered from cohorts of hospitalized patients, HCWs, and MP, before the widespread availability of COVID-19 vaccines in Serbia, we can conclude that predictors of positive PCR SARS-CoV-2 test results in MP and HCWs were similar. Accurate estimates of COVID-19 in different population groups are important for health authorities.

## 1. Introduction

Severe acute respiratory syndrome coronavirus 2 (SARS-CoV-2) and its resulting coronavirus disease 2019 (COVID-19) has caused a fast-moving pandemic [1]. Diagnostic testing, aimed to identify patients infected with SARS-CoV-2, plays a key role to control the COVID-19 pandemic [2]. All over the world, public health officials are monitoring the COVID-19 outbreak. Accurate estimates of COVID-19 in different population groups are important for the health authorities. 

During the COVID-19 pandemic, various elements contribute to the problems identified in the population of in-hospital patients, including overcrowding in hospital rooms; high rates of chronic illnesses; a great number of older patients; and shortages of physicians, nurses, and other healthcare workers (HCWs) [3]. It is not surprising that HCWs, due to the exposure to the disease and physical proximity, are at the highest risk of COVID-19 across occupational groups [4]. It is also important to estimate the burden and impact of SARS-CoV-2 on the military population (MP), since this can influence military readiness [5,6].

Hospitalized patients are considered to be more susceptible to infection because of the health conditions that caused hospitalization, and the health condition could compromise their immunity. HCWs are more exposed to the source of infection because they have contact with a great number of patients; on the other hand, they are obligated to use personal protective equipment (PPE) during those interactions. MP is a representative sample of the general adult population and referral point to compare with the other two groups. In addition to the difference in the susceptibility of the host, a difference in exposure to sources of infection, a difference in the availability and possibility of using PPE, a difference in mobility, and a difference in the types of work-related activities performed daily in the mentioned cohorts are also expected.

The lack of systematic testing across the world limits the accuracy of epidemiological data for the distribution of COVID-19 patients, especially at the beginning of the outbreak. This study aimed to investigate predictors associated with a positive polymerase chain reaction (PCR) SARS-CoV-2 test result in three cohorts: hospitalized patients, HCWs, and MP, during 2020.

## 2. Materials and Methods

Study design and settings. This retrospective cohort study was conducted at the Military Health Department (MHD), a military institution under the jurisdiction of the Ministry of Defence of the Republic of Serbia. The Military Medical Academy (MMA), Belgrade, the largest health institution in the MHD, is a 1000-bed tertiary healthcare center consisting of 27 clinics with four intensive care units, 17 institutes, and 5 centers. MMA is a very complex health institution, which also includes a Medical Faculty and employs more than 3000 people. During the COVID-19 outbreak, due to the central ventilation system, the MMA was designated to provide all specialist health services, except for COVID-19 patients. In addition to caring for members of the military system, the MMA is taking care of insured civilians during the COVID-19 pandemic.

Study population. The study kept track of the three cohorts. Respondents of the first cohort were patients hospitalized in the MMA. Respondents of the second cohort were HCWs of MMA. Respondents of the third cohort were MP (military and civilians employed in the Serbian Army and the Ministry of Defence). Through regular hospital surveillance for healthcare-associated infection, we prospectively identified patients and HCWs with symptoms of COVID-19 or contact with known COVID-19 cases within 14 days during the study period. Members of the MP were also covered by the surveillance of COVID-19 during the study period. The respondents presented to the MMA for SARS-CoV-2 testing between 25 February and 28 December 2020.

We defined a person with a positive PCR result as an individual who tested with a PCR test result for SARS-CoV-2 performed on oro- and nasopharyngeal swabs and/or on respiratory-tract secretions and aspirates. The presence of SARS-CoV-2 was detected by use of the RT-PCR method at the National referent Laboratory “Torlak Institute of Virology, Vaccines, and Serums”.

Persons with positive test results were compared with persons with negative test results in three cohorts during the study period.

Epidemiological data were collected by ten epidemiologists/preventive medicine doctors (V.Š., S.L., J.M., M.K., S.M., N.Č., I.J., J.M., S.M., A.D.) from MMA. All of them used structured questionnaires that were identical for all respondents during the collection of samples for PCR testing. Doctors asked questions and filled out the questionnaires. Epidemiological data on the following variables were gathered: demographic data (sex, age), exposure risk factors (history of travel in a country with confirmed virus transmission, treatment in a hospital with COVID-19 cases, contact with a known COVID-19 case within the previous 14 days), clinical signs and symptoms (fever, sore throat, cough, headache, myalgia/arthralgia, fatigue, gastrointestinal symptoms (nausea/vomiting/diarrhea), and pneumonia (chest X-rays or computed tomography)), and data about comorbidities (no chronic diseases, chronic cardiac disease, cardiomyopathies, hypertension, chronic pulmonary diseases, chronic liver disease, diabetes mellitus, neurological diseases, malignancy, immunodeficiency, chronic kidney disease). The medical technician (IN) entered the collected data daily into a specially created access database. The collected data were analyzed retrospectively.

Ethics approval and consent to participate. This work was completed as part of the COVID-19 outbreak operational evaluation. PCR testing was conducted without written informed consent in accordance with the recommendations of the Ministry of Health of the Republic of Serbia and the National Institute of Public Health of the Republic of Serbia [7]. During epidemiologic data collection, all respondents were orally informed about PCR testing and the purpose of data collection, and they provided oral consent. The study was officially approved by The Ethics Committee of Military Medical Academy (N28/2021, 3 December 2021). The study was conducted in compliance with recognized international standards and principles of the Declaration of Helsinki.

Statistical methods. Statistical analysis was conducted using the statistical software package IBM SPSS Statistics ver. 26.0^®^ (SPSS (Hong Kong) Ltd., Hong Kong, China). All variables were presented as frequencies of certain categories or median with IQR (interquartile range (25–75th percentile)). We used the Kolmogorov–Smirnov test for the assessment of the normality of data distribution. Calculations of odds ratios and 95% confidence intervals were performed to determine the strength of the association between risk factors and outcomes. For that purpose, the most promising independent variables as a single risk factor (unadjusted) and taken together (adjusted) were incorporated into binary logistic regression analyses. We used univariate logistic regression analysis (ULRA) and multivariate logistic regression analysis (MLRA) to identify factors associated with positive PCR SARS-CoV-2 test results among the subjects. Factors were selected with a backward stepwise method. The process involved the calculation of *p*-values for each predictor in the full model, followed by the removal of the predictor with the highest *p*-value and re-running the model until only the most significant predictors remained. Unstandardized regression coefficients (β) and odds ratios (ORs) and their 95% confidence intervals (CIs) were used to quantify the associations between variables and COVID-19 infection. The statistical significance level was set at *p* < 0.05. The relative risk (RR) was calculated by dividing the probability of an event occurring for a group with a positive PCR SARS-CoV-2 test result divided by the probability of an event occurring for a group with a negative PCR SARS-CoV-2 test result. RR values were accompanied by their 95% CI. RR value was considered clinically significant if it was less than 0.50 or more than 2.00 and the 95% CI did not include 1.

## 3. Results

A total of 6912 respondents were tested during the study period, and 1334 (19.3%) of them had a positive PCR SARS CoV-2 test result. In the cohort of patients, the median age was 68 years (54.0–76.0), and 1200 (62.9%) were male. In the cohort of HCWs, the median age was 45.0 years (36.0–54.0), and 351 (33.9%) were male. In the cohort of MP, the median age was 41.0 years (29.0–51.0), and 2890 (72.8%) were male.

Figure 1 shows the COVID-19 epidemic curve with the number of cases plotted in three cohorts by the date of diagnosis from February 25 to 28 December 2020. The peaks in the number of positive PCR test results in the cohort of patients occurred at the same time as peaks in the cohort of HCWs. 

Among 1908 patients tested, 199 (10.4%) had positive PCR tests results. If we analyzed exposure risk factors in a cohort of patients, those who tested positive for SARS-CoV-2 did not show a significantly higher rate of history travel in a country with confirmed virus transmission (1.0% vs. 0.2%; *p*: 0.092), but did show experience of treatment in a hospital with COVID-19 cases (20.1% vs. 9.0%; *p* < 0.001) and contact with known COVID-19 cases within 14 days (26.1% vs. 10.3%; *p* < 0.001) at a significantly higher frequency than those with negative tests. Those patients who tested positive for SARS-CoV-2 did not show significant differences in frequency of comorbidities (except for neurological diseases at 4.0% vs. 10.8%, *p* = 0.004) but did show some symptoms with a significantly higher frequency than those with negative tests: fever (58.3% vs. 40.3%; *p* < 0.001), sore throat (10.6% vs. 6.6%; *p* = 0.041), headache (16.6% vs. 10.3%; *p* = 0.008), and pneumonia (25.6 vs. 16.6; *p* = 0.002).

In the cohort of patients, the RR was clinically significant in exposure risk factors and some comorbidities. Namely, patients with a history of travel had more than a three times higher risk of having a positive test compared to patients with a negative test (RR: 3.22; 95% CI: 1.03–10.06). Moreover, patients with a history of treatment in hospital with COVID-19 cases (RR: 2.23; 95% CI: 1.63–3.06) and contact with a known COVID-19 case within 14 days had a more than two times higher risk of having a positive test compared to patients with a negative test (RR: 2.60; 95% CI: 1.96–3.46). Further, neurological diseases (RR: 0.37; 95% CI: 0.19–0.74) reduced the risk of positive tests by almost 40%. In the context of RR, other variables in the patient cohort did not show clinical significance.

The characteristics of the cohort of patients and related risk factors according to ULRA and MLRA are shown in Table 1. The MLRA identified four independent predictors associated with positive PCR SARS-CoV-2 test results in the cohort of patients: contact with a known COVID-19 case within 14 days (*p* < 0.001; OR: 2.56; 95% CI: 1.71–3.83), fever (*p* < 0.001; OR: 1.89; 95% CI: 1.38–2.59), pneumonia (*p* = 0.041; OR: 1.45; 95% CI: 1.01–2.09), and neurological diseases (*p* = 0.009; OR: 0.375; 95% CI: 0.18–0.78).

Among 1036 HCWs tested, 165 (15.9%) had positive PCR test results. If we analyzed exposure risk factors in a cohort of HCWs, those who tested positive for SARS-CoV-2 did not show a significantly higher rate of history of travel in a country with confirmed virus transmission (0.6% vs. 1.0%; *p* = 0.611), of experience of treatment in a hospital with COVID-19 cases (30.9% vs. 35.7%; *p* = 0.237), and contact with a known COVID-19 case within 14 days (65.5% vs. 62.3%; *p* = 0.448) in comparison with those with negative tests.

Those HCWs who tested positive for SARS-CoV-2 did not show significant differences in frequency of comorbidities but did show some symptoms with a significantly higher frequency than those with negative tests: fever (50.9% vs. 16.9%; *p* < 0.001), sore throat (38.2% vs. 23.1%; *p* < 0.001), cough (46.1% vs. 18.5%; *p* < 0.001), headache (57.6% vs. 27.1%; *p* < 0.001), myalgia/arthralgia (53.3% vs. 22.5%; *p* < 0.001), and gastrointestinal symptoms (3.0% vs. 0.3%; *p* = 0.003).

In the cohort of HCWs, the RR was clinically significant for clinical signs and symptoms. Namely, cough (RR: 2.88; 95% CI: 2.20–3.77), headache (RR: 2.89; 95% CI: 2.18–3.82, and myalgia/arthralgia (RR: 3.03; 95 CI%: 2.30–3.98) were almost three times more frequent in HCWs with a positive test compared to HCWs with a negative test. Moreover, fever (RR: 3.61; 95% CI: 2.76–4.72) and gastrointestinal symptoms (RR: 4.02; 95% CI: 2.30–7.00) were almost four times more frequent in HCWs with a positive test compared to HCWs with a negative test. In the context of RR, other variables in the HCW cohort did not show clinical significance.

The characteristics of the cohort of HCWs and related risk factors according to ULRA and MLRA are shown in Table 2. The MLRA identified four independent predictors associated with positive PCR SARS-CoV-2 test result in the cohort of HCWs: fever (*p* < 0.001; OR: 2.75; 95% CI: 1.83–4.13), cough (*p* < 0.001; OR: 2.04; 95% CI: 1.32–3.13), headache (*p* = 0.008; OR: 1.76; 95% CI: 1.15–2.68), and myalgia/arthralgia (*p* = 0.039; OR: 1.58; 95% CI: 1.02–2.45).

Among the 3968 MP tested, 970 (24.4%) had positive PCR tests. Those MP who tested positive for SARS-CoV-2 did not show significant differences in frequency of comorbidities (except for hypertension, 14.9% vs. 11.4%; *p*: 0.003, and diabetes mellitus, 4.5% vs. 3.2%; *p*: 0.045) but did show some symptoms with a significantly higher frequency than those with negative tests: fever (69.1% vs. 24.4%; *p* < 0.001), sore throat (38.0% vs. 19.2%; *p* < 0.001), cough (51.6% vs. 20.5%; *p* < 0.001), headache (53.5% vs. 23.7%; *p*: 0.028), and myalgia/arthralgia (55.2% vs. 20.0%; *p* < 0.001).

In the cohort of MP, the RR was clinically significant for clinical signs and symptoms. Namely, cough (RR: 2.74; 95% CI: 2.46–3.04), headache (RR: 2.56; 95% CI: 2.30–3.85), and myalgia/arthralgia (RR: 3.08; 95 CI%: 2.76–3.42) were almost three times more frequent in MP with a positive test compared to MP with a negative test. Moreover, fever (RR: 3.61; 95% CI: 2.76–4.72) was four times more frequent in MP with a positive test compared to MP with a negative test. In the context of RR, other variables in the MP cohort did not show clinical significance.

The characteristics of the cohort of MP and related risk factors according ULRA and MLRA are shown in Table 3. In this cohort, MLRA identified five independent predictors associated with positive PCR SARS-CoV-2 test result: contact with a known COVID-19 case within 14 days (*p* < 0.001; OR: 1.48; 95% CI: 1.25–1.76), fever (*p* < 0.001; OR: 3.66; 95% CI: 3.04–4.41), cough (*p* < 0.001; OR: 1.91; 95% CI: 1.59–2.30), headache (*p* = 0.028; OR: 1.24; 95% CI: 1.02–1.50), and myalgia/arthralgia (*p* < 0.001; OR: 1.99; 95% CI: 1.65–2.42).

## 4. Discussion

In the Republic of Serbia, the first case of COVID-19 was reported on 6 March 2020, and the outbreak is still ongoing [8]. During the period from 15 March to 7 May 2020, the government of the Republic of Serbia imposed a state of emergency due to a coronavirus pandemic across the entire country [9]. In addition, to ensure timely and coordinated treatment and to protect the citizens of Serbia, the COVID-19 crisis response team was formed in March 2020. The activities at the MHD and MMA, who were in charge of the healthcare of the military and civilian population, followed all recommendations of the COVID-19 crisis response team. Up to this day, daily testing and the monitoring of the epidemiological situation on the ground are the most important activities as the basis for adjusting preventive measures against COVID-19. Testing for COVID-19 provides us with information for a better understanding of the pandemic and the risks it poses to different populations in Serbia. This study aimed to investigate predictors associated with a positive PCR SARS-CoV-2 test in three cohorts: hospitalized patients, HCWs, and MP. Our analysis included 6912 respondents from three cohorts and used data collected during 2020, before the widespread availability of COVID-19 vaccines.

During 2 months of surveillance of COVID-19 in MP of Bolivia, 1261 or 2.5% of the MP were diagnosed [5]. During 10 months of our investigation among 3968 MP tested, 970 (24.4%) had positive SARS-CoV-2 PCR test results. Both militaries had similar challenges at the beginning of the epidemic because they helped the sanitary control of disease, especially in rural areas and at borders, but also because MP were the main force controlling quarantine blocking points for inland transportation across the countries.

Between 9 March and 15 April 2020, the occupational health service of a Massachusetts community healthcare system that had implemented a staff “hotline” system to maintain a viable/healthy workforce and among 592 HCWs tested detected 14% with positive SARS-CoV-2 assay [10], which is comparable to our results in a cohort of HCWs (15.9%). 

Taking all three cohorts together, the lowest cumulative incidence was observed among hospitalized patients, which is to be expected (10.4%). Throughout 2020, the MMA was a hospital, at times the only one in the capital, which was taking care of acute and chronic patients who did not have COVID-19. During the admission of patients for hospitalization in MMA, a serious medical triage was performed by competent clinicians. The medical triage was based on information obtained from the patients’ initial interview and examination. In case they suspected a potential COVID-19 patient, on the basis of the clinical picture (fever, shortness of breath, chest pain, dry cough, changes in senses of taste and smell, stomach problems), a rapid serological test for COVID-19 was required, as well as an X-ray of the lungs and heart and laboratory analysis (complete blood count, sedimentation, lactate dehydrogenase, C-reactive protein, ferritin, D-dimer) [11]. Although huge efforts have been made to prevent and control COVID-19, several community-acquired but also healthcare-associated infections in hospitalized patients have been registered. 

The first clinical reports from China indicated the presence of gender differences, with a predominance of male COVID-19 patients [12]. These results are in contrast to data from South Korea, which reported that 62.3% of COVID-19 patients were female [13]. As the COVID-19 pandemic spreads, the literature suggests that men tend to have a higher risk of severe infection and mortality related to COVID-19 than females [14,15,16]. In our investigation, females were dominant in the cohort of HCWs (63.0% with and 66.7% without COVID-19), and males in the cohorts of patients (65.8% with and 62.6% without COVID-19) and MP (74.0% with and 72.4% without COVID-19). This gender distribution concerning the profession is in line with the traditional role of gender in society in Serbia and the Balkans [17,18]. Our results showed no significant sex differences in susceptibility to COVID-19 through all three investigated cohorts.

Davies et al., modelled available data from China, Italy, Japan, Singapore, Canada, and South Korea and estimated that susceptibility to SARS-CoV-2 in individuals under the age of 20 is approximately half that of adults aged over 20 years and that clinical symptoms manifest in 21% of infections in 10- to 19-year-olds, rising to 69% in people aged over 70 years [19]. In 2020, no statistically significant difference in age was found between the PCR-positive and -negative respondents in our cohorts, although the cohort of patients was significantly older than the cohort of HCWs and MP. 

The role of travel in spreading infections has always been interesting. The model of Chinazzi et al., indicated that, despite the strong restrictions on traveling to and from mainland China, since January 2020, many individuals exposed to SARS-CoV-2 traveled internationally, without being detected. They also concluded that travel restrictions to COVID-19-affected areas would have modest effects, but early detection, hand washing, self-isolation, and household quarantine would be more effective in mitigating this pandemic than travel restrictions [20]. History of travel in a country with confirmed virus transmission did not influence susceptibility to COVID-19 in our study cohorts.

According to the World Health Organization, a contact is defined as anyone who has been exposed to a COVID-19 case in the following ways, within a period of 2 days before to 14 days after the case’s onset of illness: more precisely being within 1 m of a COVID-19 case for more than 15 min, or being in direct physical contact with a COVID-19 case, or providing direct care for COVID-19 patients without using PPE [21]. In our study, the MLRA identified contact with known COVID-19 cases within 14 days as an independent predictor associated with a positive PCR SARS-CoV-2 test result in a cohort of patients and MP and the family members of MP. 

The study conducted in the general population in Catalonia reported that those who tested positive for SARS-CoV-2 showed symptoms with a significantly higher frequency than those with negative tests: fever, cough, diarrhea, slurred speech, a general feeling of being unwell, an altered mental state, and anosmia [22]. Similar findings were reported by Lan et al., who recognized anosmia/ageusia, fever, and myalgia as the strongest independent predictors of a positive PCR test in a cohort of HCWs [10]. Our study found that fever, cough, headache, and myalgia/arthralgia were independently associated with positive PCR test results in a cohort of HCW.

According to previous research on the military population, epidemics of asymptomatic and mild symptomatic infections were described [23,24], except in the veteran population, in which 67.4% of patients needed hospital treatment due to the severity of COVID-19 infection [25]. In our cohort of MP, fever, myalgia/arthralgia, and cough were independent predictors of positive PCR test results.

Pneumonia complicates a modest, yet significant, proportion of patients with SARS-CoV-2 infection [26]. Throughout the analyses of our cohorts, we noticed that pneumonia was most commonly reported in a cohort of hospitalized patients and was significantly more frequent in those who tested positive for SARS-CoV-2. In addition to fever, pneumonia was an independent predictor of a positive PCR test result in the cohort of hospitalized patients.

A large meta-analysis that included a large number of observational studies conducted on a population of HCWs showed that 18.4% of the infected HCWs had pre-existing conditions, while hypertension was deemed to be the most prevalent (2.5%) [27]. Our HCWs had no chronic disease in 69.7% of infected respondents, while hypertension was reported in 16.4% of infected HCWs. A novel platform used in the military hospitals in Korea confirmed that in the ULRA, hypertension (hazard ratio—3.792) was a risk factor for earlier oxygen supplement in patients with COVID-19 admitted to two military hospitals and two civilian hospitals [28]. In our cohort of MP, hypertension was associated with a positive PCR test result in ULRA but it did not retain significance as an independent risk factor in the MLRA. It is interesting to point out that Serbia belongs to countries with a high prevalence of hypertension. Almost half of the population over the age of 15 in Serbia suffers from hypertension [29], while the prevalence of hypertension in MP over the age of 40 is almost 30% [30]. 

At the beginning of the epidemic, Oxley et al., showed that COVID-19 has been associated with an increased incidence of thrombotic events, including severe cerebrovascular events in young patients [31]. Therefore, during the surveillance of hospitalized patients, our attention was particularly focused on those with neurological diseases. Against all odds, neurological diseases were statistically more frequently registered in our hospitalized patients with negative PCR tests, which was a consequence of selection bias. A study published by Taquet et al., showed evidence for substantial neurological and psychiatric morbidity in the 6 months after COVID-19 infection [32]. Scientists are still trying to figure out how the virus affects the brain and other parts of the central nervous system.

This study had several limitations. First, due to the lack of a reference diagnostic test for COVID-19, a small number of positive and negative respondents could have been misclassified on account of the false-positive or false-negative results. Another major limitation was the lack of genomic characterization of SARS-CoV-2, which circulated in our cohorts during the study period. The data presented here were obtained during 2020, the first year of the pandemic, before the introduction of the vaccination program against COVID-19 in Serbia, which can be considered as a strength rather than a limitation of the study because the impact of the vaccine on infection was avoided. However, the strengths of our study included its cohort design. Furthermore, data were collected by epidemiologists/preventive medicine doctors of MMA, which strengthened the accuracy of the cohort design. Another upside of the study was the sample size. Finally, to our knowledge, for the first time, three different populations were monitored at the same time.

## 5. Conclusions

According to data gathered from cohorts of hospitalized patients, HCWs, and MP, before the widespread availability of COVID-19 vaccines in Serbia, we can conclude that predictors of positive PCR test results in MP and HCWs were similar, namely, contact with known cases within 14 days, cough, headache, and myalgia/arthralgia were independently associated with positive PCR SARS CoV-2 test results in a cohort of MP. Furthermore, fever, cough, headache, and myalgia/arthralgia were independently associated with positive PCR SARS-CoV-2 test results in a cohort HCWs. Moreover, independent predictors of positive PCR SARS-CoV-2 test results in hospitalized patients were contact with known cases within 14 days, fever, and pneumonia.

## Figures and Tables

**Figure 1 ijerph-20-03601-f001:**
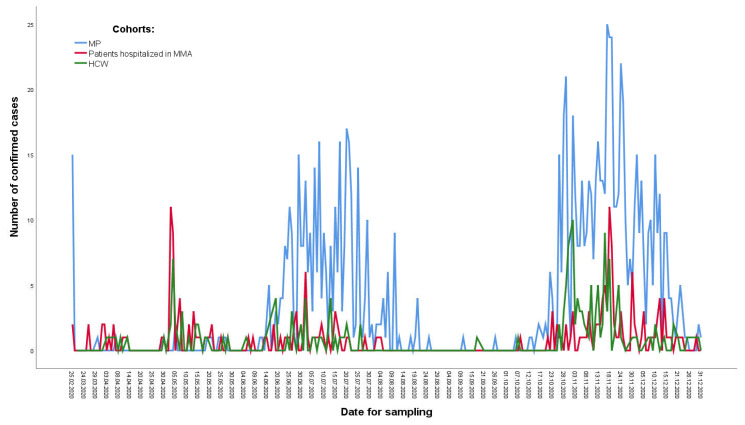
Distribution of COVID-19 in three cohorts during 2020.

**Table 1 ijerph-20-03601-t001:** The characteristics of the cohort of patients and related risk factors according ULRA * and MLRA ^†^.

	Respondents with Positive Test (n = 199) (%)	Respondents with Negative Test (n = 1709) (%)	RR ^§^ (95% CI)	ULRA * Crude OR ^‡^ (95%CI)	ULRA * *p*	MLRA ^†^ Adjusted (95%CI)	MLRA ^†^ *p*
**Demographics**							
Male, n (%) Female, n (%)	131 (65.8) 68 (34.2)	1069 (62.6) 640 (37.4)	10.9/9.6 = 1.14 (0.86–1.50)	0.86 (0.64–1.18)	0.365	/	/
Age, median (IQR)	67 (53–75)	68 (55–76)		0.99 (0.99–1.00)	0.224	/	/
**Exposure risk factors**							
History of travel in a country with confirmed virus transmission, n (%)	2 (1.0)	4 (0.2)	33.1/10.4 = 3.22 (1.03–10.06)	4.33 (0.79–23.78)	0.092	/	/
Treatment in a hospital with COVID-19 cases, n (%)	40 (20.1)	153 (9.0)	20.7/9.3 = 2.23 (1.63–3.06)	2.56 (1.74–3.76)	<0.001	1.51 (0.97–2.36)	0.067
Contact with known COVID-19 cases within 14 days, n (%)	52 (26.1)	176 (10.3)	22.8/8.8 = 2.60 (1.96–3.46)	3.08 (2.17–4.38)	<0.001	2.56 (1.71–3.83)	<0.001
**Clinical signs and symptoms**							
Fever, n (%)	116 (58.3)	688 (40.3)	14.4/7.5 = 1.92 (1.47–2.51)	2.07 (1.54–2.79)	<0.001	1.89 (1.38–2.59)	<0.001
Sore throat, n (%)	21 (10.6)	113 (6.6)	15.7/10.0 = 1.57 (1.03–2.37)	1.67 (1.02–2.72)	0.041	1.11 (0.65–1.88)	0.697
Cough, n (%)	56 (28.1)	409 (23.9)	12.0/9.9 = 1.21 (0.91–1.62)	1.245 (0.90–1.73)	0.191	/	/
Headache, n (%)	33 (16.6)	176 (10.3)	15.8/9.8 = 1.61 (1.14–2.28)	1.73 (1.15–2.60)	0.008	1.35 (0.87–2.09)	0.176
Myalgia/arthralgia, n (%)	28 (14.1)	220 (12.9)	11.3/10.3 = 1.10 (0.75–1.60)	1.11 (0.72–1.69)	0.635	/	
Fatigue, n (%)	1 (0.5)	4 (0.2)	20.0/10.4 = 1.92 (0.33–11.15)	2.15 (0.24–19.36)	0.494	/	/
Gastrointestinal symptoms, n (%)	1 (0.5)	18 (1.1)	5.3/10.5 = 0.50 (0.07–3.40)	0.47 (0.06–3.57)	0.469	/	/
Pneumonia, n (%)	51 (25.6)	284 (16.6)	15.2/9.4 = 1.62 (1.20–2.17)	1.73 (1.23–2.44)	0.002	1.46 (1.02–2.09)	0.041
**Comorbidities**							
No chronic diseases, n (%)	29 (14.6)	209 (12.2)	12.2/10.2 = 1.20 (0.83–1.73)	1.22 (0.805–1.86)	0.344	/	/
Chronic cardiac disease, n (%)	53 (26.6)	433 (25.5)	10.9/10.3 = 1.06 (0.79–1.43)	1.07 (0.77–1.49)	0.691	/	/
Cardiomyopathy, n (%)	8 (4.0)	95 (5.6)	7.8/10.6 = 0.73 (0.37–1.45)	0.71 (0.34–1.49)	0.363	/	/
Hypertension, n (%)	77 (38.7)	701 (41.0)	9.9/10.8 = 0.92 (0.70–1.20)	0.91 (0.67–1.23)	0.528	/	/
Chronic pulmonary diseases, n (%)	22 (11.1)	192 (11.2)	10.3/10.4 = 0.98 (0.65–1.50)	0.98 (0.61–1.57)	0.940	/	/
Chronic liver diseases, n (%)	9 (4.5)	52 (3.0)	14.8/10.3 = 1.44 (0.77–2.66)	1.51 (0.73–3.11)	0.265	/	/
Diabetes mellitus, n (%)	36 (18.1)	310 (18.1)	10.4/10.4 = 1.00 (0.71–1.40)	0.99 (0.68–1.46)	0.987	/	/
Neurological diseases, n (%)	8 (4.0)	185 (10.8)	4.1/11.1 = 0.37 (0.19–0.74)	0.34 (0.17–0.71)	0.004	0.37 (0.18–078)	0.009
Malignancy, n (%)	52 (26.1)	395 (23.1)	11.6/10.1 = 1.15 (0.86–1.56)	1.18 (0.84–1.65)	0.342	/	/
Immunodeficiency, n (%)	4 (2.0)	32 (1.9)	11.1/10.4 = 1.07 (0.42–2.71)	1.07 (0.38–3.07)	0.893	/	/
Chronic kidney disease, n (%)	18 (9.0)	186 (10.9)	8.8/10.6 = 0.83 (0.52–1.32)	0.81 (0.49–1.35)	0.428	/	/

* ULRA—univariate logistic regression analysis. ^†^ MLRA—multivariate logistic regression analysis. ^§^ RR—relative risk. ^‡^ OR—odds ratio.

**Table 2 ijerph-20-03601-t002:** The characteristics of the cohort of HCWs and related risk factors according ULRA * and MLRA ^†^.

Characteristics	Respondents with Positive Test (n = 165) (%)	Respondents with Negative Test (n = 871) (%)	RR § (95% CI)	ULRA Crude OR ‡ (95%CI)	ULRA *p*	MLRA Adjusted OR ‡ (95%CI)	MLRA *p*
**Demographics**							
Male, n (%) Female, n (%)	61 104	290 581	17.4/15.2 = 1.14 (0.86–1.53)	0.85 (0.60–1.20)	0.361	/	/
Age, median (IQR)	44 (36–53)	45 (36–54)	/	0.99 (0.98–1.01)	0.618	/	
**Exposures risk factors**							
History of travel in a country with confirmed virus transmission, n (%)	1 (0.6)	9 (1.0)	10.0/16.0 = 0.62 (0.10–4.04)	0.58 (0.07–4.64)	0.611	/	/
Treatment in a hospital with COVID-19 cases, n (%)	51 (30.9)	311 (35.7)	14.1/16.9 = 0.83 (0.61–1.13)	0.81 (0.56–1.15)	0.237	/	/
Contact with a known COVID-19 case within 14 days, n (%)	108 (65.5)	543 (62.3)	16.6/14.8 = 1.12 (0.83–1.50)	1.14 (0.81–1.62)	0.448	/	/
**Clinical signs and symptoms**							
Fever, n (%)	84 (50.9)	147 (16.9)	36.4/10.1 = 3.61 (2.76–4.72)	5.11 (3.59–7.27)	<0.001	2.75 (1.83–4.13)	<0.001
Sore throat, n (%)	63 (38.2)	201 (23.1)	23.9/13.2 = 1.81 (1.36–2.39)	2.06 (1.45–2.93)	<0.001	0.86 (0.56–1.33)	0.499
Cough, n (%)	76 (46.1)	161 (18.5)	32.1/11.1 = 2.88 (2.20–3.77)	3.766 (2.65–5.35)	0.001	2.04 (1.32–3.14)	0.001
Headache, n (%)	95 (57.6)	236 (27.1)	28.7/9.9 = 2.89 (2.18–3.82)	3.65 (2.59–5.15)	<0.001	1.76 (1.15–2.68)	0.008
Myalgia/arthralgia, n (%)	88 (53.3)	196 (22.5)	31.0/10.2 = 3.03 (2.30–3.98)	3.94 (2.79–5.56)	<0.001	1.58 (1.02–2.45)	0.039
Fatigue, n (%)	2 (1.2)	8 (0.9)	20.0/15.9 = 1.26 (0.36–4.38)	1.32 (0.28–6.29)	0.724	/	/
Gastrointestinal symptoms, n (%)	5 (3.0)	3 (0.3)	62.5/15.6 = 4.02 (2.30–7.00)	9.04 (2.14–38.21)	0.003	3.38 (0.69–16.44)	0.132
Pneumonia, n (%)	2 (1.2)	5 (0.6)	28.6/15.8 = 1.80 (0.55–5.87)	2.14 (0.41–11.05)	0.370	/	/
**Comorbidities**							
No chronic diseases, n (%)	115 (69.7)	555 (63.7)	17.2/13.7 = 1.26 (0.92–1.71)	1.31 (0.91–1.88)	0.142	/	/
Chronic cardiac disease, n (%)	5 (3.0)	28 (3.2)	15.2/16.0 = 0.95 (0.42–2.16)	0.94 (0.36–2.47)	0.902	/	/
Cardiomyopathy, n (%)	/	/	0.0/15.9 = /	/	/	/	/
Hypertension, n (%)	27 (16.4)	135 (15.5)	16.7/15.8 = 1.06 (0.72–1.54)	1.07 (0.68–1.67)	0.779	/	/
Chronic pulmonary diseases, n (%)	1 (0.6)	18 (2.1)	5.3/16.1 = 0.33 (0.05–2.21)	0.29 (0.04–2.18)	0.229	/	/
Chronic liver diseases, n (%)	/	1 (0.1)	0.0/15.9 = /	/	1.000	/	/
Diabetes mellitus, n (%)	4 (2.4)	28 (3.2)	12.5/16.0 = 0.78 (0.31–1.97)	0.75 (0.26–2.16)	0.592	/	/
Neurological diseases, n (%)	1 (0.6)	/	100.0/15.8 = 6.31 (5.48–7.26)	/	1.000	/	/
Malignancy, n (%)	1 (0.6)	9 (1.0)	10.0/16.0 = 0.63 (0.10–4.04)	0.58 (0.07–4.64)	0.611	/	/
Immunodeficiency, n (%)	/	/	0.0/15.9 = /	/	1.000	/	/
Chronic kidney disease, n (%)	/	1	0.0/15.9 = /	/	1.000	/	/

* ULRA—univariate logistic regression analysis. ^†^ MLRA—multivariate logistic regression analysis. ^§^ RR—relative risk. ^‡^ OR—odds ratio.

**Table 3 ijerph-20-03601-t003:** The characteristics of the cohort of MP and related risk factors according ULRA * and MLRA ^†^.

Characteristics	Respondents with Positive Test (n = 970) (%)	Respondents with Negative Test (n = 2998) (%)	RR ^§^ (95% CI)	ULRA Crude OR ^‡^ (95% CI)	ULRA *p*	MLRA Adjusted OR ^‡^ (95% CI)	MLRA *p*
**Demographics**							
Male, n (%) Female, n (%)	718 252	2172 826	24.8/23.4 = 1.06 (0.94–1.20)	0.92 (0.78–1.09)	0.339	/	/
Age, median (IQR)	40 (27–51)	41 (29–51)	/	0.99 (0.99–1.01)	0.182	/	/
**Exposures risk factors**							
History of travel in a country with confirmed virus transmission, n (%)	9 (0.9)	39 (1.3)	18.8/24.5 = 0.77 (0.42–1.38)	0.71 (0.34–1.47)	0.358	/	/
Treatment in a hospital with COVID-19 cases, n (%)	/	/	/	/	/	/	/
Contact with a known COVID-19 case within 14 days, n (%)	457 (47.1)	886 (29.6)	34.0/19.5 = 1.74 (1.56–1.94)	2.12 (1.83–2.46)	<0.001	1.48 (1.25–1.76)	<0.001
**Clinical signs and symptoms**							
Fever, n (%)	670 (69.1)	733 (24.4)	47.8/11.7 = 4.08 (3.62–4.60)	6.90 (5.88–8.09)	<0.001	3.66 (3.04–4.41)	<0.001
Sore throat, n (%)	369 (38.0)	576 (19.2)	39.0/19.9 = 1.96 (1.76–2.19)	2.58 (2.20–3.02)	<0.001	0.93 (0.76–1.13)	0.443
Cough, n (%)	501 (51.6)	614 (20.5)	44.9/16.4 = 2.74 (2.46–3.04)	4.15 (3.56–4.84)	<0.001	1.91 (1.59–2.30)	<0.001
Headache, n (%)	519 (53.5)	712 (23.7)	42.2/16.5 = 2.56 (2.30–2.85)	3.70 (3.17–4.30)	<0.001	1.24 (1.023–1.50)	0.028
Myalgia/arthralgia, n (%)	535 (55.2)	599 (20.0)	47.2/15.3 = 3.08 (2.76–3.42)	4.93 (4.22–5.75)	<0.001	2.00 (1.65–2.42)	<0.001
Fatigue, n (%)	3 (0.3)	21 (0.7)	12.5/24.5 = 0.51 (0.18–1.47)	0.44 (0.13–1.48)	0.184	/	/
Gastrointestinal symptoms, n (%)	14 (1.4)	34 (1.1)	29.2/24.4 = 1.20 (0.77–1.86)	1.28 (0.68–2.39)	0.445	/	/
Pneumonia, n (%)	6 (0.6)	22 (0.7)	21.4/24.5 = 0.88 (0.43–1.78)	0.84 (0.34–2.08)	0.710	/	/
**Comorbidities**							
No chronic diseases, n (%)	693 (71.4)	2158 (72.0)	24.3/24.8 = 0.98 (0.87–1.11)	0.97 (0.83–1.14)	0.746	/	/
Chronic cardiac disease, n (%)	28 (2.9)	67 (2.2)	29.5/24.3 = 1.21 (0.88–1.66)	1.30 (0.83–2.03)	0.250	/	/
Cardiomyopathies, n (%)	/	/	0.0/24.5 = /	/	/		/
Hypertension, n (%)	145 (14.9)	341 (11.4)	29.8/23.7 = 1.26 (1.08–1.46)	1.37 (1.11–1.69)	0.003	1.12 (0.88–1.44)	0.355
Chronic pulmonary diseases, n (%)	13 (1.3)	27 (0.9)	32.5/24.4 = 1.33 (0.85–2.09)	1.49 (0.77–2.91)	0.237	/	/
Chronic liver disease, n (%)	/	2 (0.1)	0.0/24.5 = /	/	0.999	/	/
Diabetes mellitus, n (%)	44 (4.5)	95 (3.2)	31.7/24.2 = 1.31 (1.02–1.68)	1.45 (1.01–2.09)	0.045	0.27 (0.83–1.95)	0.268
Neurological diseases, n (%)	4 (0.4)	7 (0.2)	36.4/24.4 = 1.49 (0.68–3.26)	1.77 (0.52–6.06)	0.363	/	/
Malignancy, n (%)	16 (1.6)	73 (2.4)	18.0/24.6 = 0.73 (0.47–1.14)	0.67 (0.39–1.16)	0.154	/	/
Immunodeficiency, n (%)	/	/	50.0/24.4 = 2.05 (0.51–8.19)	3.09 (0.19–49.49)	0.425	/	/
Chronic kidney disease, n (%)	2 (0.2)	14 (0.5)	12.5/24.5 = 0.51 (0.14–1.87)	0.44 (0.10–1.94)	0.279	/	/

* ULRA—univariate logistic regression analysis. ^†^ MLRA—multivariate logistic regression analysis. ^§^ RR—relative risk. ^‡^ OR—odds ratio.

## Data Availability

The datasets used and/or analyzed in the present study are available from the corresponding author upon reasonable request.

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
