# Peer review of "Epidemiological Predictors of Positive SARS-CoV-2 Polymerase Chain Reaction Test in Three Cohorts: Hospitalized Patients, Healthcare Workers, and Military Population, Serbia, 2020"

_ijerph, 2023, doi:10.3390/ijerph20043601_

Round 1

Reviewer 1 Report

I would like to commend the authors for this prospective study initiated in the very beginning of pandemic. Authors studied the epidemiological predictors of COVID-19 in individuals with symptoms typical for COVID-19. The study showed that epidemiological predictors were similar between three cohorts: hospitalized patients, healthcare workers (HCW), and military personnel (MP). The study showed that contact with known cases within 14 days, cough, headache, and myalgia/arthralgia were independently associated with COVID-19 in MP cohort; fever, cough, headache, and myalgia/arthralgia were independently associated with COVID-19 in HCW cohort; and contact with known cases within 14 days, fever and pneumonia were independently associated with COVID-19 in hospitalized patients.

The title of paper begins with "Epidemiological predictors of COVID-19 in three cohorts…". In my opinion, it seems that it would be more accurate to say "Epidemiological predictors of positive COVID-19 test results in three cohorts…". The term "epidemiological predictors" belongs to the context of predictive medicine and predictive models, which allow to identify people or populations at elevated disease risk whereas risk is something that increases the chance of developing a disease. Comorbidities, age, and other clinical and demographic characteristics indeed can be true predictors of disease, but fever, cough, headache, myalgia/arthralgia, etc. being signs of manifested infection are rather predictors of positive COVID-19 test result because the disease had already occurred.

The authors state that "Epidemiological data were collected by epidemiologist/preventive medicine doctors from MMA". It is not clear how these data were collected. Did epidemiologist or preventive medicine doctors use any questionnaire? If yes, was the questionnaire filled out by respondents or by doctors? Was the questionnaire identical for all cohorts? Did authors use data from electronic medical records or from any medical information system to collect epidemiological data? How many doctors were involved in collecting epidemiological data? Authors are encouraged to describe the procedure of data collection in more detail.

Authors write that "This study aimed to investigate differences in predictors associated 167 with COVID-19 in three cohorts: hospitalized patients, HCWs and MP." Indeed, authors investigated the differences in predictors and signs of COVID-19 between symptomatic and asymptomatic individuals who tested positive and negative for COVID-19 in the three cohorts. However, I do not see much of statistical analysis of differences in predictors between three mentioned cohorts. If authors aimed at comparing predictors between the cohorts (hospitalized patients, HCW, and MP), then matching of cohorts perhaps with calculation of propensity score would be required for such an inter-cohort comparison. An opinion of professional statistician would help in this regard.

Author Response

I would like to commend the authors for this prospective study initiated in the very beginning of pandemic. Authors studied the epidemiological predictors of COVID-19 in individuals with symptoms typical for COVID-19. The study showed that epidemiological predictors were similar between three cohorts: hospitalized patients, healthcare workers (HCW), and military personnel (MP). The study showed that contact with known cases within 14 days, cough, headache, and myalgia/arthralgia were independently associated with COVID-19 in MP cohort; fever, cough, headache, and myalgia/arthralgia were independently associated with COVID-19 in HCW cohort; and contact with known cases within 14 days, fever and pneumonia were independently associated with COVID-19 in hospitalized patients.

Response: Thank you for giving us the opportunity to submit a revised draft of our manuscript. We appreciate the time and effort that you dedicated to providing feedback on our manuscript and are grateful for the insightful comments on and valuable improvements to our paper. We have been able to incorporate changes to reflect most of the suggestions. We marked all changes in the text of the manuscript in red. You can track changes according to your requirements.

The title of paper begins with "Epidemiological predictors of COVID-19 in three cohorts…". In my opinion, it seems that it would be more accurate to say "Epidemiological predictors of positive COVID-19 test results in three cohorts…". The term "epidemiological predictors" belongs to the context of predictive medicine and predictive models, which allow to identify people or populations at elevated disease risk whereas risk is something that increases the chance of developing a disease. Comorbidities, age, and other clinical and demographic characteristics indeed can be true predictors of disease, but fever, cough, headache, myalgia/arthralgia, etc. being signs of manifested infection are rather predictors of positive COVID-19 test result because the disease had already occurred.

Response: We thank Reviewer 1 for this valuable comment. As the reviewer advised, in the title and the entire text of the paper, we changed "predictors of COVID-19" to “predictors of positive PCR SARS CoV-2 test results”

The authors state that "Epidemiological data were collected by epidemiologist/preventive medicine doctors from MMA". It is not clear how these data were collected. Did epidemiologist or preventive medicine doctors use any questionnaire? If yes, was the questionnaire filled out by respondents or by doctors? Was the questionnaire identical for all cohorts? Did authors use data from electronic medical records or from any medical information system to collect epidemiological data? How many doctors were involved in collecting epidemiological data? Authors are encouraged to describe the procedure of data collection in more detail.

Response: Thanks for this comment. We have described in detail the procedure of data collection and their entry into the database.

“Epidemiological data were collected by ten epidemiologists/preventive medicine doctors (VŠ, SL, JM, MK, SM, NČ, IJ, JM, SM, AD) from MMA. All of them used structured questionnaire identical to all respondents, during the collection of samples for PCR testing. Doctors were asking questions and filled out the questionnaires. Epidemiological data on the following variables were gathered: demographic data (sex, age), exposure risk factors (history of travel in country with confirmed virus transmission, treatment in hospital with COVID-19 cases, contact with known COVID-19 case within previous 14 days), clinical signs and symptoms - fever, sore throat, cough, headache, myalgia/arthralgia, fatigue, gastrointestinal symptoms (nausea/ vomiting/diarrhea), pneumonia (chest X-rays or computed tomography) and data about comorbidities (no chronic diseases, chronic cardiac dis-ease, cardiomyopathies, hypertension, chronic pulmonary diseases, chronic liver disease, diabetes mellitus, neurological diseases, malignancy, immunodeficiency, chronic kidney disease). The medical technician (IN) daily entered the collected data into a specially created Access database. The collected data were analyzed retrospectively”

Authors write that "This study aimed to investigate differences in predictors associated 167 with COVID-19 in three cohorts: hospitalized patients, HCWs and MP." Indeed, authors investigated the differences in predictors and signs of COVID-19 between symptomatic and asymptomatic individuals who tested positive and negative for COVID-19 in the three cohorts. However, I do not see much of statistical analysis of differences in predictors between three mentioned cohorts. If authors aimed at comparing predictors between the cohorts (hospitalized patients, HCW, and MP), then matching of cohorts perhaps with calculation of propensity score would be required for such an inter-cohort comparison. An opinion of professional statistician would help in this regard.

Response: We thank the Reviewer 1 for raising this critical issue. We agree that the differences between the three cohorts were not clearly highlighted. That was the reason we contacted a professional statistician who pointed out the following. Propensity score matching is a tool for causal inference in nonrandomized studies that allows for conditioning on large sets of covariates. The propensity score is defined as the probability of receiving treatment based on measured covariates. The balance that a randomized experiment is expected to create by design is here established through statistical matching.

Propensity score matching consists of several analytic steps:

Researchers select a set of pre-test covariates that are deemed important based on theoretical arguments. This step is critical as the credibility of the propensity score analysis hinges on the selection of proper covariates. Covariates of convenience (e.g. gender, age) are usually not sufficient instead researchers should strive to build a convincing case that no unobserved confounders are omitted. Based on this set of covariates the propensity score is estimated. This is often done using logistic regression in which the treatment assignment is used as the outcome variable and the selected covariates as predictors. After the estimation of the propensity score, the actual matching procedure commences. After matching is completed, a series of model adequacy checks should be performed. The main interest of the researcher is to check whether the balance on the covariates has truly been achieved through the matching procedure. In the last step, the treatment effect is estimated in the matched subsample.

The main problem in applying this statistical method to our respondents is already in the first step, which is the selection of covariables on the basis of which the respondents should be matched. We are not sure how they would match respondents in three cohorts, because all the monitored variables are too significant to be examined as risk factors for contracting COVID-19. Therefore, we believe that this statistical technique is not the most suitable for our work. Do you have any other suggestions? Or to change the research objective, if you agree “This study aimed to investigate differences in predictors associated with positive polymerase chain reaction (PCR) SARS CoV-2 test in three cohorts: hospitalized patients, HCWs, and MP, during 2020.” Of course, we have to change the Title - “Epidemiological predictors of positive SARS CoV-2 polymerase chain reaction test in three cohorts: hospitalized patients, healthcare workers, and military population, Serbia, 2020. Are there any differences?

Reviewer 2 Report

- The Results section of the abstract must be described in greater detail.

- The Introduction section of the manuscript is poorly described. The authors must introduce why they think there could be differences between the three groups.

- In the Ethics paragraph, the author stated "PCR testing was conducted without written informed consent in accordance with the recommendations of the Ministry of Health of the Republic of Serbia and the National Institute of Public Health of the Republic of Serbia.". Please give relevant evidence of this issue, indicating also a reference.

- The authors stated this is a prospective cohort study conducted in 2020. However, the study was approved by The Ethics Committee of Military Medical Academy in 2021 (N28/2021, 3rd December 2021). How is this possible?

- From a methologogical point of view, how did the authors collect data for calculating the follow-up period for each participant?

- The multivariate analysis chosen was a logistic regression analysis, that calculates Odds ratios and not Relative risks. Moreover, the authors carried out a cohort study and it shold be more appropriate to conduct a Poisson regression or a Cox model, that are capable to calculate relative risks.

- The procedure of the backward procedure must be presented with more details.

Author Response

Thank you for giving us the opportunity to submit a revised draft of our manuscript. We appreciate the time and effort that you dedicated to providing feedback on our manuscript and are grateful for the insightful comments on and valuable improvements to our paper. We have been able to incorporate changes to reflect most of the suggestions. We marked all changes in the text of the manuscript in red. You can track changes according to your requirements.

- The Results section of the abstract must be described in greater detail.

Response: We thank Reviewer 2 for raising this critical issue. We have described the Results section of the abstract in greater detail. Also, we have described the Results section in greater detail. The only reason for the modest presentation of the results section of the abstract is the limited number of words required by the Journal Instruction for Authors for manuscript preparation (“The abstract should be a total of about 200 words maximum”)

- The Introduction section of the manuscript is poorly described. The authors must introduce why they think there could be differences between the three groups.

Response: We thank Reviewer 2 for this valuable comment. We specified in the introduction why we think that there could be differences between the three groups.

“Hospitalized patients are considered to be more susceptible to infection because of health conditions which caused hospitalization and that health condition could compromise their immunity. HCWs are more exposed to the source of infection because they have contact with a great number of patients; on the other hand, they are obligated to use PPE during those interactions. MP is a representative sample of the general adult population and referral point to compare with the other two groups. In addition to the difference in the susceptibility of the host, a difference in exposure to sources of infection, a difference in the availability and possibility of using personal protective equipment, a difference in mobility, a difference in the types of work-related activities performed daily in the mentioned cohorts is also expected.”

- In the Ethics paragraph, the author stated "PCR testing was conducted without written informed consent in accordance with the recommendations of the Ministry of Health of the Republic of Serbia and the National Institute of Public Health of the Republic of Serbia.". Please give relevant evidence of this issue, indicating also a reference.

Response: We thank Reviewer 2 for raising this critical issue. We have added relevant evidence of this issue, reference N° 7

“This work was completed as part of the COVID-19 outbreak operational evaluation. PCR testing was conducted without written informed consent in accordance with the recommendations of the Ministry of Health of the Republic of Serbia and the National Institute of Public Health of the Republic of Serbia [7]. During epidemiologic data collection, all respondents were orally informed about PCR testing and the purpose of data collection and they gave oral consent. The study was officially approved by The Ethics Committee of the Military Medical Academy (N28/2021, 3rd December 2021).

Reference 7 : Professional and methodological guidelines to control the introduction and prevention of the spread of the new corona virus sars-cov-2 in the Republic of Serbia Available online: http://demo.paragraf.rs/demo/combined/Old/t/t2020_04/PP_004_2020_002.htm (accessed on 28 December 2022).

- The authors stated this is a prospective cohort study conducted in 2020. However, the study was approved by The Ethics Committee of the Military Medical Academy in 2021 (N28/2021, 3rd December 2021). How is this possible?

Response: Thanks for this comment. We have tried to specify our activities.”During epidemiologic data collection, all respondents were orally informed about PCR testing and the purpose of data collection and they gave oral consent.” The study was officially approved aftermath by The Ethics Committee of Military Medical Academy (N28/2021, 3rd December 2021). Also, we have changed the design of the study to a retrospective cohort study.

- From a methodological point of view, how did the authors collect data for calculating the follow-up period for each participant?

Response: We thank Reviewer 2 for raising this critical issue. We have added a follow-up period in the part Study population

“Through regular hospital surveillance for healthcare-associated infection, we prospectively identified patients and HCWs with symptoms of COVID-19 or contact with known COVID-19 cases within 14 days during the study period. Members of the MP were also covered by the surveillance of COVID-19 during the study period. The respondents presented to the MMA for SARS-CoV-2 testing between 25th February and 28th December 2020.”

- The multivariate analysis chosen was a logistic regression analysis, that calculates Odds ratios and not Relative risks. Moreover, the authors carried out a cohort study and it should be more appropriate to conduct a Poisson regression or a Cox model, that are capable to calculate relative risks.

Response: We thank the Reviewer for the comment.  We expressed the odds ratio as you suggested and additionally calculated the relative risk, which we inserted into the tables for each cohort.

- The procedure of the backward procedure must be presented with more details.

Response: We thank the Reviewer for the comment. We have included an explanation of the backward procedure with more details in the part Statistical methods “The process involved the calculation of p-values for each predictor in the full model, followed by the removal of the predictor with the highest p-value and re-running the model until only the most significant predictors remain.”

Round 2

Reviewer 2 Report

In the tables and in the text report the relative risks with 95% confidence intervals

Author Response

In the tables and in the text, we have presented relative risks with
95% confidence intervals. Changes are marked in yellow.